# Non-response in a national health survey in Germany: An intersectionality-informed multilevel analysis of individual heterogeneity and discriminatory accuracy

**Philipp Jaehn**[1]*, **Emily Mena**[2,3], **Sibille Merz**[1], **Robert Hoffmann**[4], **Antje Gößwald**[4], **Alexander Rommel**[4], **Christine Holmberg**[1,5], on behalf of the ADVANCE GENDER study group[¶]

**1** Institute of Social Medicine and Epidemiology, Brandenburg Medical School Theodor Fontane, Brandenburg an der Havel, Germany, **2** Department of Social Epidemiology, Institute of Public Health and Nursing Research, University of Bremen, Bremen, Germany, **3** Health Sciences Bremen, University of Bremen, Bremen, Germany, **4** Department of Epidemiology and Health Monitoring, Robert Koch-Institute, Berlin, Germany, **5** Faculty of Health Sciences, Joint Faculty of the Brandenburg University of Technology Cottbus–Senftenberg, the Brandenburg Medical School Theodor Fontane and the University of Potsdam, Brandenburg an der Havel, Germany

¶ Membership of the ADVANCE GENDER study group is listed in the Acknowledgments.
* philipp.jaehn@mhb-fontane.de

**Data Availability Statement:** The data from the DEGS1 study cannot be made publicly available because informed consent from study participants

## Abstract

### Background

Dimensions of social location such as socioeconomic position or sex/gender are often associated with low response rates in epidemiological studies. We applied an intersectionality-informed approach to analyze non-response among population strata defined by combinations of multiple dimensions of social location and subjective health in a health survey in Germany.

### Methods

We used data from the cross-sectional sample of the German Health Interview and Examination Survey for Adults (DEGS1) conducted between 2008 and 2011. Information about non-responders was available from a mailed non-responder questionnaire. Intersectional strata were constructed by combining all categories of age, sex/gender, marital status, and level of education in scenario 1. Subjective health was additionally used to construct intersectional strata in scenario 2. We applied multilevel analysis of individual heterogeneity and discriminatory accuracy (MAIHDA) to calculate measures of discriminatory accuracy, proportions of non-responders among intersectional strata, as well as stratum-specific total interaction effects (intersectional effects). Markov chain Monte Carlo methods were used to estimate multilevel logistic regression models.

### Results

Data was available for 6,534 individuals of whom 36% were non-responders. In scenario 2, we found weak discriminatory accuracy (variance partition coefficient = 3.6%) of

did not cover public deposition of data and publicly providing an anonymized version of the analytical data set used in our current analysis would not comply with current data protection regulations in Germany as anonymized information could still be used in combination and/or with other data to identify DEGS1 study participants. However, the minimal data set underlying the findings presented in this article is archived in the 'Health Monitoring' Research Data Centre at the Robert Koch Institute (RKI). The 'Health Monitoring' Research Data Centre is accredited by the German Data Forum according to uniform and transparent standards. Access to a Scientific Use File of the minimal data set is possible. Scientific Use Files need to be requested at the RKI´s 'Health Monitoring' Research Data Centre (e-mail: fdz@rki.de).

**Funding:** This study was funded by the German Ministry of Education and Research (URL: https://www.bmbf.de/, grant number 01GL1710A, PI: Christine Holmberg). The funder was not involved in study design, data collection and analysis, decision to publish, or preparation of the manuscript.

**Competing interests:** The authors have declared that no competing interests exist.

intersectional strata, while predicted proportions of non-response ranged from 20.6% (95% credible interval (CI) 17.0%-24.9%) to 57.5% (95% CI 48.8%-66.5%) among intersectional strata. No evidence for intersectional effects was found. These results did not differ substantially between scenarios 1 and 2.

## Conclusions

MAIHDA revealed that proportions of non-response varied widely between intersectional strata. However, poor discriminatory accuracy of intersectional strata and no evidence for intersectional effects indicate that there is no justification to exclusively target specific intersectional strata in order to increase response, but that a combination of targeted and population-based measures might be appropriate to achieve more equal representation.

## Introduction

Representativeness is a crucial element of external validity of epidemiological studies, especially if studies aim at estimating disease frequency or measures of population impact within specific societies [1, 2]. Descriptive comparisons of the study population with the target population aid the identification of population groups for which research results might be less representative. Current guidelines in epidemiology suggest the comparison of social, demographic and health-related features between study and target population [1, 3, 4]. We argue that a critical reflection of participants' social location from an intersectional perspective could serve as an important framework for describing representativeness of population-based studies. Social location has been defined as an individual's location in relations of hierarchy and power at a specific time and in a specific social context [5]. Dimensions of social location are inter alia socioeconomic position, race/ethnicity, sex/gender, place of residence, religion, social capital, age, sexuality or (dis)ability [5, 6].

The concept of intersectionality was developed by Black feminist scholars during the 1980s and 1990s [5]. Unique lived realities and experiences of discrimination against African-American women that were incomparable to either discrimination against white women or Black men were used as a starting point to develop the theoretical framework [7]. Intersectionality posits that social systems of power are mutually constituting and reinforcing [8]. An intercategorical intersectional perspective focuses, thereby, on combinations of multiple dimensions of social location [9]. These combinations, or intersectional strata, are conceptualised to be incomparable to one another [10]. Thus, intersectionality expands traditional analytic frameworks of epidemiology that focus either on single dimensions of social location or on mutually adjusted associations [11]. Besides social location, intersectionality may focus on social identity or on social processes such as discrimination [11].

Study participation has been suggested to be context-specific [1, 12, 13]. However, it is feasible to draw a summary of dimensions of social location that have been associated with study participation across different contexts and study designs. In population-based surveys, ethnic minority status, low income, low level of education and young and old age were associated with lower response proportions [14–16]. Furthermore, male sex, low socioeconomic status, being an unskilled worker, having no children and being unmarried were associated with low study participation in cross-sectional and cohort designs [13, 17–19].

Finally, associations of single dimensions of social location with study participation might vary across categories of further dimensions of social location. In a population-based cohort

study in Germany, for example, men living in a steady partnership showed a higher response rate compared to men not living in a steady partnership, while response rates among women did not differ according to partnership status [20]. Study participation was comprehensively examined in the GAZEL study, an occupational cohort study conducted in France [13]. In GAZEL, a sampling frame with information about several social and demographic characteristics was available for all non-participants [13]. The authors found good evidence for interaction of employment grade with year of birth, level of education and number of children when investigating study participation [13]. This suggests that an intersectionality-informed perspective on study participation might be informative to uncover differential patterns of response and to mirror social complexity in such analyses [5, 10].

Finally, intersectional Multilevel Analysis of Individual Heterogeneity and Discriminatory Accuracy (MAIHDA) has recently been developed to inform quantitative data analysis by an intercategorical intersectional framework [21–23]. Main goals of MAIHDA are to estimate measures of discriminatory accuracy, measures of disease frequency, and stratum-specific total interaction effects (so-called intersectional effects) when using multiple dimensions of social location to build intersectional strata [21, 24, 25]. Evaluations of this method regarding complementarity with intersectional theory as well as statistical properties have been published recently [24–28].

In this study, we applied MAIHDA to investigate the association of intersectional strata with non-response using data of a large cross-sectional health interview and examination survey in Germany. We operationalised intersectional strata as combinations of age, sex/gender, marital status, and level of education (scenario 1). Besides social location, health status is an important predictor of study participation [16, 18, 19, 29]. Therefore, we chose to include subjective health in addition to dimensions of social location in scenario 2 of MAIHDA.

## Materials and methods

### Study design and population

We used data of the cross-sectional German Health Interview and Examination Survey for Adults (DEGS1) carried out 2008 through 2011 [30–32]. The DEGS1 sample consisted of participants of the German National Health Interview and Examination Survey (GNHIES98) conducted in 1998 and an extension sample that was drawn from population registers [32–34]. We only included participants of the extension sample and excluded all individuals who participated in GNHIES98 (N = 4,055), because we aimed at estimating study participation in a sample that most likely had not had prior experience with health research.

DEGS1 followed a two-stage stratified cluster sampling design. The target population with respect to cross-sectional analyses were residents of Germany aged 18 to 79 years. In the first stage of the sampling strategy, 180 communities were sampled from a sampling frame of all communities in Germany stratified by federal state and type of community. Community types in Germany are classified according to population density, grade of urbanisation and administrative borders [35]. Of those communities, 120 were sampled for GNHIES98 and 60 were additionally drawn for DEGS1 using probability proportional to community size sampling in both situations. These communities represented the primary sampling units (PSUs). Within each PSU, simple random samples of individuals were drawn from population registers stratified by 10-year age groups. As the study population of GNHIES98 grew approximately 10 years older until DEGS1, a new cross-sectional sample was drawn in the youngest 10-year age group. The study design of DEGS1 is described in detail elsewhere [30, 31].

Overall, 11,008 people were invited to participate in DEGS1 for the first time. 4,193 participated, 5,754 did choose not to participate (non-responders), and the remainder were quality neutral losses. 2,342 of all 5,754 non-responders (42%) responded to a non-responder survey

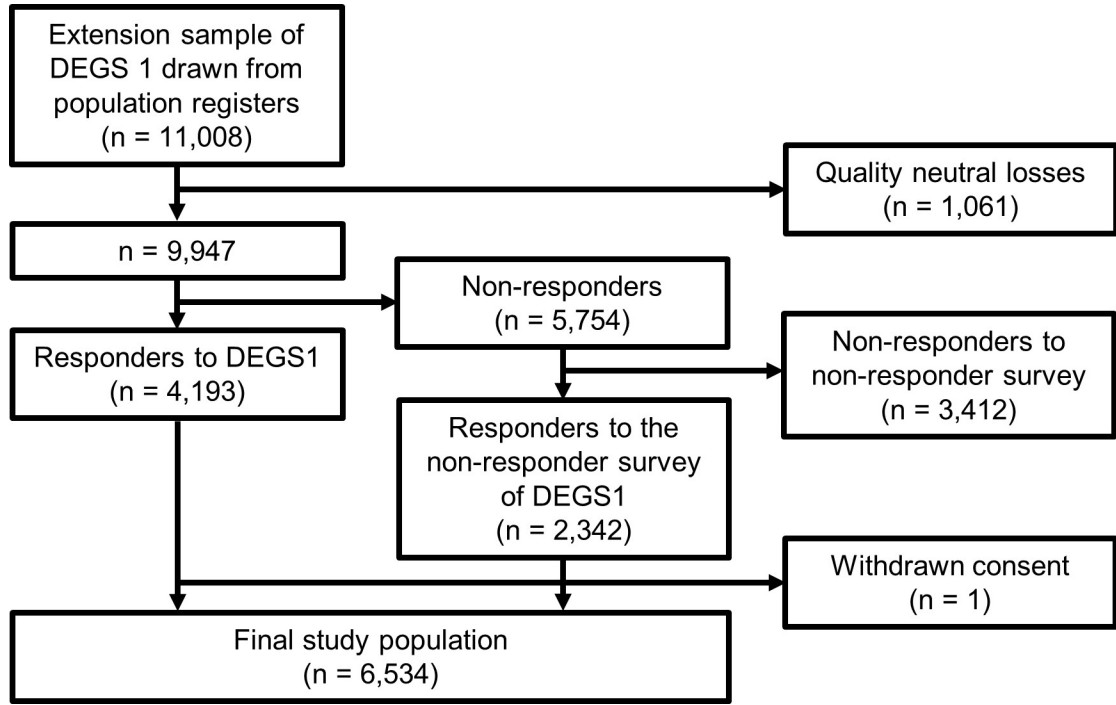

**Fig 1. Study population flowchart.**

[32]. One person, who participated initially in DEGS1 withdrew consent. Correspondingly, our final sample consisted of 6,534 people (Fig 1).

The purpose of DEGS1 was to obtain representative estimates of disease burden, prevalence of diseases and risk factors, disability and health care utilisation for the German population. During fieldwork, trained health professionals visited study sites successively. Information about non-responders was obtained from a non-responder survey, conducted by mail, telephone and in person during (failed) participant recruitment. Interview data in this study were collected using self-administered questionnaires and computer-assisted questionnaires (CAPI). Questionnaires were available in German, Russian, Turkish, Serbo-Croatian and English [30–32].

### Variables

Being a non-responder in the sample of DEGS1 was defined as the outcome of interest. Non-responders were individuals who completed the non-responder survey but did not participate in the DEGS1 study. Responders were individuals who visited one of the study centres to participate in DEGS1 [32].

The studied dimensions of social location were age, sex/gender, marital status, and level of education. Age was classified into three age groups (18–39, 40–59 and 60–79 years). For all other variables binary categories were chosen. Categories of sex/gender were "female" and "male". Having no degree of secondary education or a grade IX lower secondary degree ("Hauptschulabschluss") were defined as "low level of education". Having any other degree was defined as "high level of education". Marital status was classified as "being married" or "not being married". Not being married included being single, divorced, or widowed. Finally, subjective health was measured on a Likert scale in five categories (very good, good, moderate, bad, very bad). The responses "very good" and "good" were categorised as "good subjective health", while the other responses were categorised as "moderate or bad subjective health".

Classifications were chosen in order to yield both meaningful categories and sufficiently large numbers of observations within each intersectional stratum. Intersectional strata were obtained by forming all possible combinations of categories of age, sex/gender, marital status and level of education (3x2x2x2 = 24) (scenario 1). These intersectional strata were additionally stratified by subjective health, rendering 48 strata (scenario 2). The inclusion of subjective health in this analysis is in line with the concept of intersectionality, since healthy bodies or psychological distress might well be related to social privilege or stigma [36–38]. As subjective health might represent a location in social power hierarchies, we would argue that an inclusion in this analysis is warranted. Additionally, subjective health has frequently been associated with study participation [16, 18, 19]. To underscore that subjective health could be a disputable dimension of social location and to estimate changes of results after the inclusion of subjective health, we conducted the two scenarios of the analysis that are described above.

## Statistical methods

We calculated descriptive statistics and univariable associations of all single dimensions of social location and subjective health with being a non-responder in DEGS1. Odds ratios, 95% confidence intervals and p-values from likelihood-ratio tests were calculated using random intercepts logistic regression with PSUs as random effects to account for clustered sampling.

MAIHDA was used to estimate proportions of non-responders within intersectional strata, measures of discriminatory accuracy and intersectional effects. MAIHDA uses a multilevel modelling approach where intersectional strata are modelled as level two random effects. The method has been originally developed for linear regression [21–23] and was subsequently applied to logistic regression [39]. We followed the approach proposed by Axelsson-Fisk et al. for multilevel logistic regression [39]. Therefore, random intercepts logistic regression models were used where individuals were cross-classified in intersectional strata and PSUs. PSUs were included to account for the clustered sampling design.

We applied two scenarios of intersectional MAIHDA. In scenario 1, we used the predefined 24 intersectional strata, which were built by age, sex/gender, marital status and level of education. In scenario 2, we used the predefined 48 intersectional strata which were built by all dimensions of social location of scenario 1 and subjective health. We used data with complete observations of all five variables that were used to build intersectional strata (N = 6,432).

Subsequently, for both scenario 1 and 2, a null model was compared to a model including fixed effects of the single variables used to construct intersectional strata [21]. To do so, we fit a null model that included a term for the intercept, no fixed effects but intersectional strata as well as PSUs as cross-classified random effects. Next, single variables used to build intersectional strata were included as fixed effects in addition to all parameters of the null model. Fixed effects in scenario 1 were age, sex/gender, marital status and level of education. Subjective health was added as fixed effect in scenario 2. These models were used to calculate four quantities of interest: (A) measures of discriminatory accuracy for intersectional strata, (B) the proportion of variance explained by adding fixed effects, (C) proportions of non-responders within intersectional strata, and (D) total interaction effects (intersectional effects) for each intersectional stratum.

To estimate measures of discriminatory accuracy of intersectional strata, variance partition coefficients (VPCs) were calculated for both null models and models including fixed effects [39]:

$$VPC = \frac{\sigma_{uj}^2}{\sigma_{uj}^2 + \sigma_{uk}^2 + \frac{\pi^2}{3}} x100\% \tag{1}$$

where $\sigma_{uj}$ represents the variance of intersectional strata and $\sigma_{uk}$ the variance of PSUs of the respective model. VPCs give the proportion of variance attributable to variance between intersectional strata and have been argued to represent discriminatory accuracy of intersectional strata [24].

To estimate the proportion of variance explained by adding fixed effects, the proportional change in variance (PCV) of intersectional strata between null model and model including fixed effects was calculated [39]:

$$PCV = \frac{\sigma^2_{uj(1)} - \sigma^2_{uj(2)}}{\sigma^2_{uj(1)}} \qquad (2)$$

where $\sigma_{uj(1)}$ represents the variance of intersectional strata in the null model and $\sigma_{uj(2)}$ represents the variance of intersectional strata in the model including fixed effects. The PCV represents the proportion of the total between-stratum variance of intersectional strata of the null model that is explained after having added all fixed effects (39). The lower the PCV, the higher the amount of "unexplained" variance, which can be due to interaction effects or to omitted variable bias [21].

Finally, the model including fixed effects was used to calculate total predicted proportions of non-responders within each intersectional stratum. Therefore, predicted log odds of non-response based on estimated fixed effects and random effects were calculated for each intersectional stratum. The predicted log odds were transformed to proportions using the inverse logit function.

To calculate intersectional effects within each intersectional stratum, predicted proportions of non-responders based on fixed effects alone were calculated for each intersectional stratum [21]. Log odds of non-response based on fixed effects were also transformed to proportions using inverse logits. Finally, intersectional effects were derived by subtracting predicted proportions of non-responders based on fixed effects alone from total predicted proportions of non-responders in both scenarios [39]. The transformation from log odds to proportions yields intersectional effects that are deviations from predictions of additive combinations of the fixed effects [39].

All MAIHDA models were estimated using Markov Chain Monte Carlo (MCMC) methods. We used the MCMCglmm package (version 2.29) in R (version 3.6.0). MCMCglmm has been shown to perform comparable to other statistical programs for cross-classified multilevel logistic regression models [40]. We used weakly informative priors and ran all analyses using 50,000 iterations with a burn-in period of 5,000 and a thinning interval of 50 iterations [39]. MCMC chains were checked graphically for convergence. Point estimates were the means of the respective MCMC chains for each parameter. 95% credible intervals (95% CI) were obtained using the 2.5[th] and 97.5[th] percentiles of the respective MCMC chains [39].

## Ethics statement

All procedures performed in DEGS1 were in accordance with the ethical standards of the ethics committee of the Charité-Universitätsmedizin Berlin (Reference No. EA2/047/08). Participants of DEGS1 provided written informed consent prior to the interview and examination. This study was part of the joint project AdvanceGender [41]. AdvanceGender was in accordance with the ethical standards of the ethics committee of the Brandenburg Medical School (Reference No. E-01-20180529). All procedures were in accordance with the 1964 Helsinki declaration and its later amendments or comparable ethical standards.

## Results

Among the total sample of DEGS1 used in this study, 36% (2342/6534) were non-responders (Table 1). 39% (2561/6534) of the sample were under the age of 40 years and 31% (2035/6534) were over 60 years. Furthermore, 51% (3356/6534) were female, 45% (2908/6479) were not married, 32% (2088/6473) had a low level of education and 26% (1716/6506) rated their health as moderate or bad. There was strong evidence for an association of age, level of education, marital status and subjective health with non-response (Table 1). Compared to the youngest age group, the odds of non-response were lower in the group of people aged 40 to 59 and higher in the group aged 60 or older. The odds of non-response were furthermore higher when being unmarried, having a low level of education and rating one's subjective health as moderate or bad compared to the respective reference groups. Odds of non-response among females were 5% lower compared to males, however, there was no evidence for an association between sex/gender and non-response.

In scenario 1 of MAIHDA, the smallest number of observations in an intersectional stratum was 66 (23 non-responders). Numbers of observations in each intersectional stratum are displayed in S1 Table. Results for fixed effects indicated elevated odds among unmarried compared to married people and among people with a low compared to people with a high level of education (Table 2). The VPC of the null model was 3.3%. The VPC dropped to 0.8% after adding the fixed effects for age, sex/gender, marital status and level of education. PCV in scenario 1 was 74.4%, indicating that 25.6% of variance was not explained by adding fixed

**Table 1. Study characteristics and univariable associations of dimensions of social location with being a non-responder.**

| | N | Non-Responder n (%) | OR[1] | 95% conf. int.[2] | p-value[3] |
|---|---|---|---|---|---|
| Age | | | | | |
| 18–39 | 2561 | 921 (36.0) | 1.00 | (ref.) | |
| 40–59 | 1938 | 636 (32.8) | 0.87 | 0.76–0.99 | |
| 60–79 | 2035 | 785 (38.6) | 1.13 | 0.99–1.27 | <0.001 |
| missing | 0 | | | | |
| Sex | | | | | |
| male | 3178 | 1162 (36.6) | 1.00 | (ref.) | |
| female | 3356 | 1180 (35.2) | 0.95 | 0.86–1.06 | 0.37 |
| missing | 0 | | | | |
| Marital status | | | | | |
| married | 3571 | 1208 (33.8) | 1.00 | (ref.) | |
| not married | 2908 | 1129 (38.8) | 1.23 | 1.11–1.37 | <0.001 |
| missing | 55 | | | | |
| Educational level | | | | | |
| high | 4385 | 1416 (32.3) | 1.00 | (ref.) | |
| low | 2088 | 911 (43.6) | 1.62 | 1.44–1.81 | <0.001 |
| missing | 61 | | | | |
| Subjective health | | | | | |
| good | 4790 | 1602 (33.4) | 1.00 | (ref.) | |
| moderate or bad | 1716 | 738 (43.0) | 1.52 | 1.35–1.70 | <0.001 |
| missing | 28 | | | | |
| total | 6534 | 2342 (35.8) | | | |

[1] odds ratio from multilevel logistic regression with PSUs as random effects

[2] 95% confidence interval

[3] p value from likelihood-ratio test

**Table 2. Fixed effects, variance partition coefficients and proportional change of variance (scenarios 1 and 2).**

| | scenario 1 | | | scenario 2 | | |
|---|---|---|---|---|---|---|
| | **OR** | | **95% CI** | **OR** | | **95% CI** |
| Age | | | | | | |
| 18–39 | 1.00 | | (ref.) | 1.00 | | (ref.) |
| 40–59 | 0.84 | | 0.64–1.09 | 0.82 | | 0.66–1.05 |
| 60–79 | 1.04 | | 0.80–1.32 | 0.93 | | 0.74–1.17 |
| Sex | | | | | | |
| Male | 1.00 | | (ref.) | 1.00 | | (ref.) |
| female | 0.92 | | 0.72–1.11 | 0.92 | | 0.75–1.08 |
| Marital status | | | | | | |
| Married | 1.00 | | (ref.) | 1.00 | | (ref.) |
| not married | 1.39 | | 1.12–1.74 | 1.37 | | 1.14–1.66 |
| Educational level | | | | | | |
| High | 1.00 | | (ref.) | 1.00 | | (ref.) |
| low | 1.76 | | 1.45–2.21 | 1.66 | | 1.36–1.97 |
| Subjective health | | | | | | |
| Good | | | | 1.00 | | (ref.) |
| moderate or bad | | | | 1.49 | | 1.20–1.77 |
| | **VPC** | (%) | **95% CI** | **VPC** | (%) | **95% CI** |
| Null model | 3.3 | | 1.5–6.7 | 3.6 | | 2.1–5.7 |
| Model including fixed effects | 0.8 | | 0.1–2.1 | 0.9 | | 0.2–2.1 |
| | **PCV** | (%) | **95% CI** | **PCV** | (%) | **95% CI** |
| | 74.4 | | 25.7–97.2 | 74.8 | | 30.3–95.7 |

OR: odds ratio

95% CI: 95% credible interval

VPC: Variance partition coefficient

PCV: Proportional change in variance from null model to the model including fixed effects

effects. The predicted proportions of non-responders ranged from 23% to 53% (Fig 2). Intersectional effects ranged from -3.0% to 4.8%, all 95% CI of intersectional effects crossed 0% (Fig 3).

After having added subjective health to build the intersectional strata (scenario 2), 3 strata had less than 20 observations, but no intersectional stratum had less than 10 observations (S2 Table). Results for fixed effects indicated evidence for elevated odds among people rating their health as moderate or bad compared to people rating their health as good (Table 2). The VPC of the null model was 3.6% and the VPC of the model including fixed effects was 0.9%. The PCV in scenario 2 was 74.8%, indicating that 25.2% of variance was not explained by adding fixed effects. Predicted proportions of non-responders ranged from 20% to 57% (Fig 4). Intersectional effects ranged from -3.6% to 6.4%, all 95% CIs of intersectional effects included 0% (Fig 5). Two intersectional strata showed large intersectional effects. Women aged 60 to 79 years who were unmarried, had low education and rated their health as moderate or bad showed an intersectional effect of 6.3% (95% CI -0.9% to 16.0%). Furthermore, men aged 18 to 39 who were unmarried, had a low level of education and rated their health as good had an intersectional effect of 6.4% (95% CI -1.3% to 15.4%). The predicted proportions of non-responders among these groups were among the highest of all intersectional strata with 51% and 57% respectively.

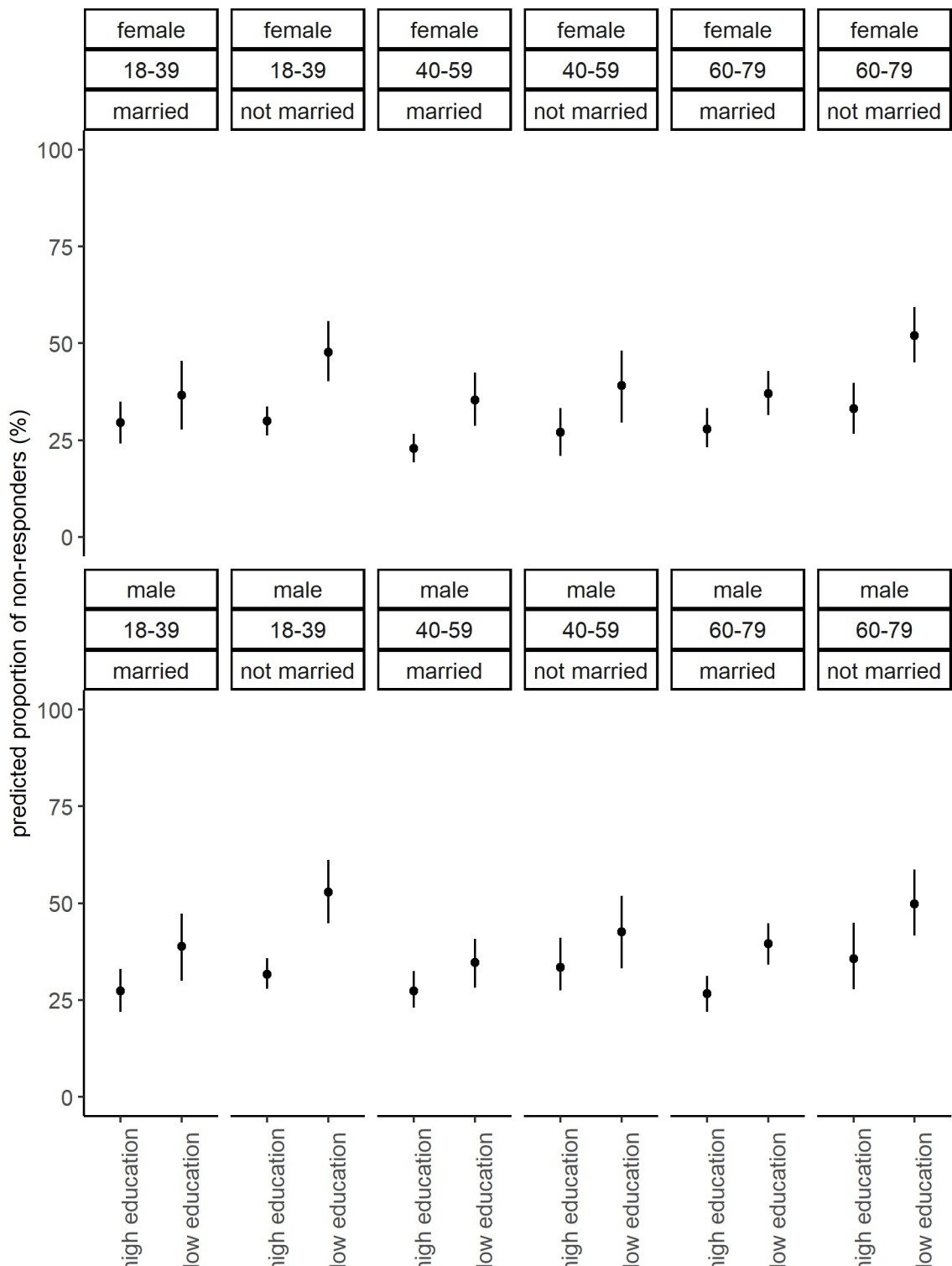

**Fig 2. Predicted proportions of non-responders for each stratum from scenario 1.** Point estimates are proportions and 95% credible intervals.

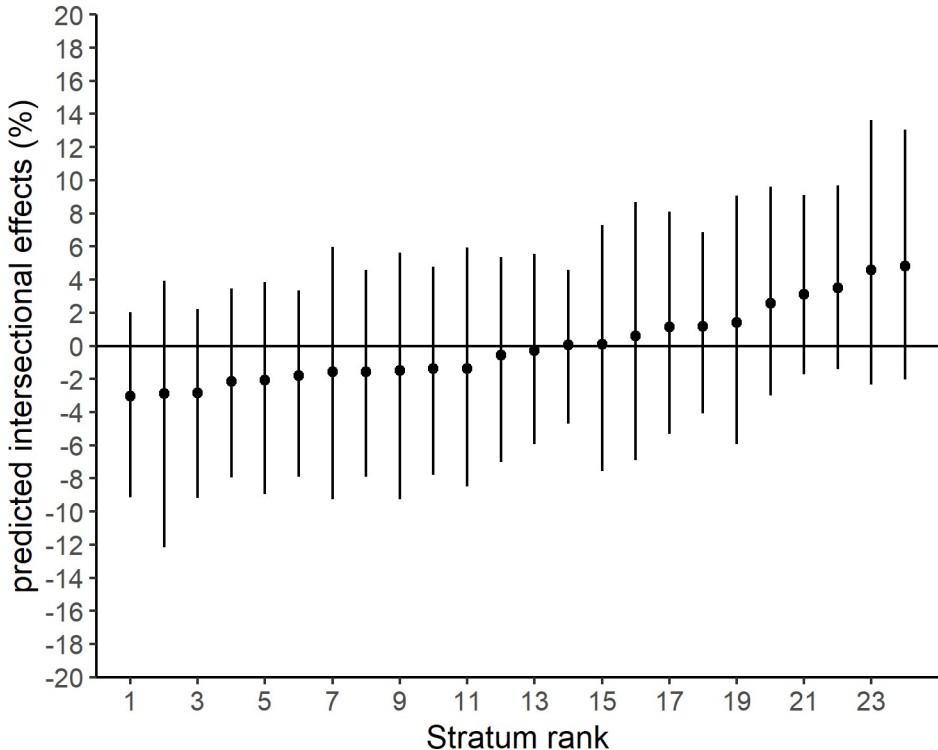

**Fig 3. Predicted intersectional effects for each stratum from scenario 1.** Point estimates are proportions and 95% credible intervals. Intersectional strata are ranked by the size of the predicted intersectional effect.

## Discussion

We investigated non-response to a population-based health survey in Germany using an intersectionality-informed approach. When using age, sex/gender, marital status, level of education and subjective health to build intersectional strata, predicted proportions of non-responders ranged from 20% to 57%. In this scenario we found a VPC of 3.6% indicating low discriminatory accuracy of intersectional strata. About 25% of the variance between intersectional strata could not be explained by adding fixed effects. Adding subjective health did not change the VPC or PCV compared to a scenario excluding subjective health. We did not find evidence for intersectional effects of any particular intersectional stratum in both scenarios as all CI of intersectional effects crossed 0. However, the intersectional effect for women aged 60 to 79 years who were unmarried, had low education and rated their health as moderate or bad and the intersectional effect for men aged 18 to 39 who were unmarried, had a low level of education and rated their health as good were relatively large with intersectional effects of 6.3% and 6.4% respectively.

42% of all non-responders participated in the non-responder survey of DEGS1 [32]. Thus, there remains uncertainty whether data gathered from non-responders is representative for all non-responders. To assess this possible selection bias, we compared proportions of sex/gender, marital status and level of education among responders in DEGS1 to census data for Germany (S1 Appendix). Compared to these census data, men, people not in marriage, and people with a low level of education are underrepresented in DEGS1. Data on subjective health was not available from the census and proportions of age groups could not be compared due to the stratified sampling of DEGS1 [30]. These crude comparisons indicate that our results are not generalizable to the German population, however, our results may allow conclusions about the directions of over- or underrepresentation.

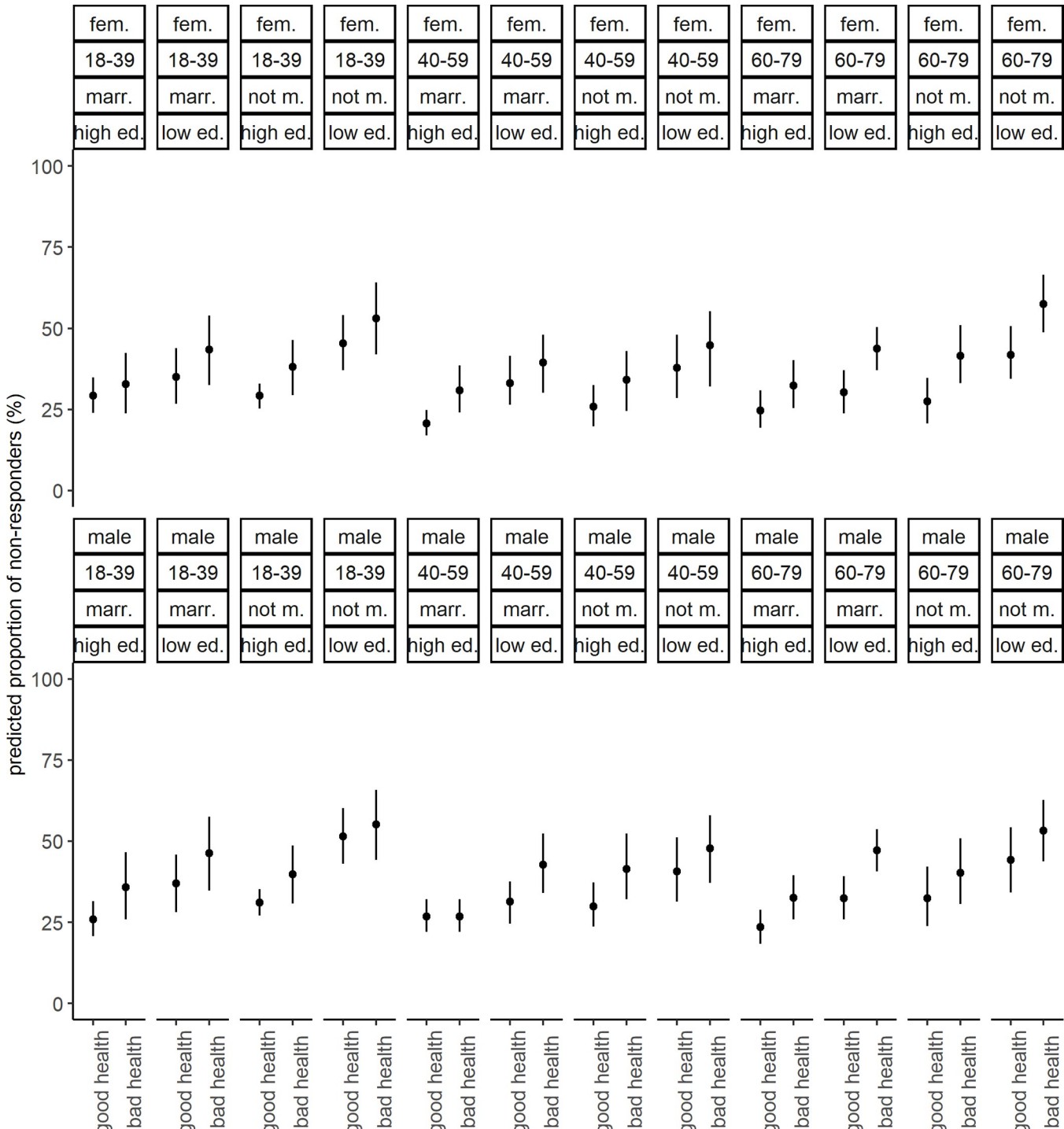

**Fig 4. Predicted proportions of non-responders for each stratum from scenario 2.** Point estimates are proportions and 95% credible intervals. marr.: married, not m.: not married, high ed.: high education, low ed.: low education, bad health: moderate or bad health.

We chose three categories for age, because previous studies suggested a low response among very young and very old people in Germany [20, 42]. The characteristics age, sex/gender and level of education were frequently used in recent quantitative intersectional analyses

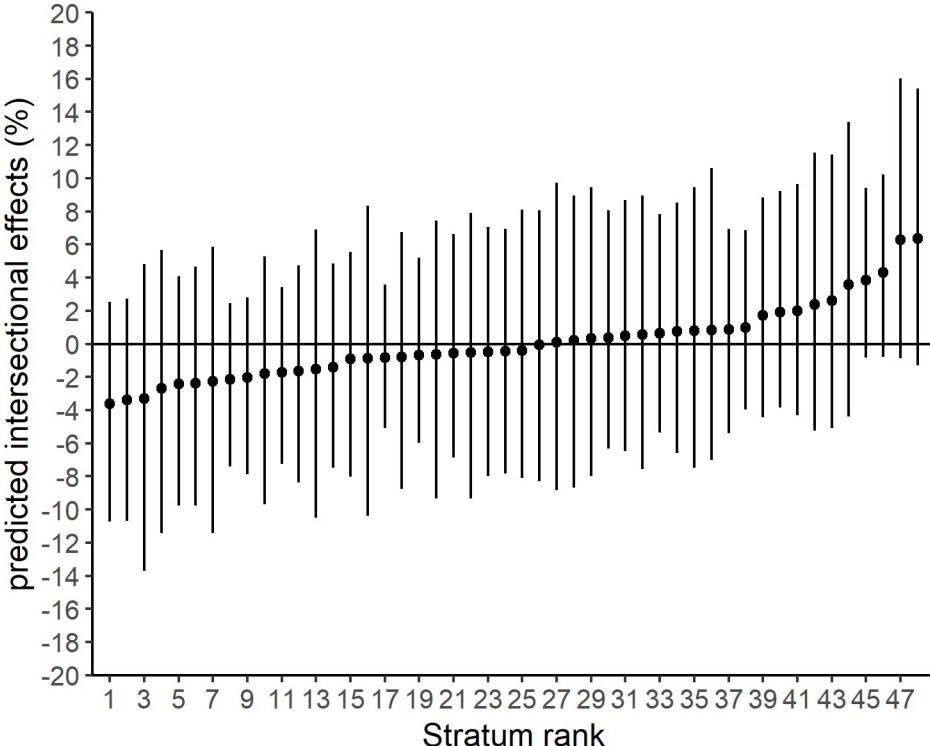

**Fig 5. Predicted intersectional effects for each stratum from scenario 2.** Point estimates are proportions and 95% credible intervals. Intersectional strata are ranked by the size of the predicted intersectional effect.

[21, 22, 39, 43–45]. Instead of marital status, previous studies used cohabitation or civil status [39, 43, 45, 46]. Both measures probably crudely operationalise social support, however, they might capture different aspects of this broad concept [47]. A further important dimension of social location in Germany is race/ethnicity or migration [48]. Information on nationality, which is not comparable to race/ethnicity, was available in the data, however, the number of people with non-German nationality in our sample (N = 516) was too low to construct intersectional strata with sufficient numbers of observations in each stratum.

Finally, subjective health has not been included in intersectional MAIHDA so far. We chose to include a measure on subjective health separately in a second scenario, because it was consistently described as a strong predictor of study participation and because interaction of subjective health with intersectional strata might be suggestive for selection bias [16, 18–20]. Furthermore, we would argue that subjective health is related to social privilege or stigma, because it measures both mental and psychological well-being [36, 37, 49, 50]. Therefore, an inclusion of subjective health in an intersectionality-informed analysis might be justified. Finally, unchanged PCVs between scenario 1 and 2 suggest that adding subjective health does not increase variance unexplained by additive fixed effects. Furthermore, no additional intersectional effects were revealed in scenario 2.

MAIHDA is a new method that is still to be tested in simulation studies and real world applications. We adopted recently proposed methods for cross-classified multilevel logistic regression in MAIHDA [22, 39]. Sample size in this study is sufficient to estimate a multilevel model since 3 of 48 strata had less than 20 observations in scenario 2 and all strata had over 30 observations in scenario 1 [51]. Nonetheless, our sample size might be at the lower boundary of scenarios in which MAIHDA is advantageous over single level models [21, 25]. We used a

cross-classified regression analysis to account for the clustered sampling design of DEGS1. Cross-classified MAIHDA could, furthermore, be used to separate geographical from socio-economic influences, which was beyond the scope of the present analysis. However, promising approaches to incorporate contextual-level determinants in MAIHDA have been published recently [44].

Compared to single-level intersectional analyses, the risk of a type one error is reduced in MAIHDA by precision-weighting of intersectional strata [26]. However, in presence of inter-action between fixed effects, precision-weighting performs less well than in regular applica-tions of multilevel modelling [26]. Furthermore, estimates of fixed effects in MAIHDA are different from fixed effects in single level models as they are derived from precision-weighted grand means instead of population means [25, 27]. A grand mean in MAIHDA is the mean of means of intersectional strata [25]. A major advantage of MAIHDA is the estimation of mea-sures of discriminatory accuracy besides traditional comparison of population averages [24]. Interpreting measures of discriminatory accuracy together with population averages enables more comprehensively informed decisions about suitable public health action to address an issue under study [24].

It has been emphasised that MAIHDA is a tool to descriptively explore or map outcomes of interest across intersectional strata, which we considered a suitable and innovative approach for our research question [21]. Crucially, MAIHDA models remain parsimonious when inves-tigating multiple combinations of social characteristics and are easier to interpret than for example single level models with multiple interaction terms [21]. Estimates of intersectional effects are displayed as deviations from predictions of additive main effects, which is in line with the proposition of intersectionality [11, 21]. Furthermore, comparisons to a master cate-gory are avoided in MAIHDA as no reference groups are used to estimate intersectional effects [21]. MAIHDA visualises combinations of privileged and oppressed social locations, which contributes to avoid essentialism and prioritising any system of oppression over the other [10]. Finally, MAIHDA is a convenient method when researchers aim to err on the conservative side [22].

To our knowledge, this is the first study to present an intersectionality-informed quantita-tive analysis of study participation. Previous research has suggested that associations of single dimensions of social location with study participation might differ when combined with fur-ther dimensions [13, 20]. An intersectionality-informed perspective might contribute to cap-ture these differential patterns of non-response more comprehensively. Results from the fixed effects of our MAIHDA models agree with previous research suggesting that people with a lower level of education, who are unmarried or report worse self-rated health show low response proportions [13, 16, 19]. Research on study participation from Germany suggested that young men and old women might be participating less frequently in epidemiological stud-ies [20, 42]. Our result of high proportions of non-responders among multiply marginalised old women and young men are in line with these findings. However, as all CIs of intersectional effects included zero, there was little statistical evidence for well-known patterns of interaction such as the interaction of sex/gender and age.

An intersectionality-informed approach to the relationship of social location with study participation opens up several perspectives that cannot be addressed by conventional analyses. First, low VPCs in our study illustrated that intersectional strata have low discriminatory accu-racy as it has been suggested that VPCs between 1% and 5% are poor [39]. Hence, these char-acteristics may not be suited to predict study participation. Low VPCs also indicate that heterogeneity within intersectional strata is high. As no strong evidence for intersectional effects could be found, our results indicate that there is no justification to exclusively target specific intersectional strata in order to increase response, but that a combination of targeted

and population-based measures might be appropriate to achieve more equal representation [24]. Finally, the total proportion of variance between intersectional strata that was not explained by fixed effects was substantial with 25%, suggesting the presence of complex inter-actions between all included variables. We presented a first attempt to capture this complexity. However, our results should be confirmed in further research before generalizable conclusions can be drawn.

## Conclusions

We need a more complex understanding of study participation having in mind the steadily decreasing participation in population-based studies and increasing diversity of contemporary societies. Selection bias occurs in descriptive epidemiology if sub-groups of a population that suffer from a high disease burden are underrepresented. This scenario might apply for popula-tions at the intersection of multiple axes of oppression, because, besides being at high risk for many diseases, marginalised groups have been excluded from health research and often choose not to participate in studies [52]. An intersectional framework might be an important first step to consider the multiplicity of systems of privilege and oppression when investigating partici-pation and representativeness of population-based studies. Our results show that MAIHDA may be suited to operationalise an intersectionality-informed analysis of non-response and we suggest to conduct further analyses in order to add evidence to this topic.

## Supporting information

**S1 Table. Number of observations within intersectional strata, predicted proportions of non-responders and intersectional effects of scenario 1.** Intersectional strata are ranked by predicted proportions of non-responders.
(DOCX)

**S2 Table. Number of observations within intersectional strata, predicted proportions of non-responders and intersectional effects of scenario 2.** Intersectional strata are ranked by predicted proportions of non-responders.
(DOCX)

**S1 Appendix. Comparison of the cross-sectional sample of DEGS1 with census data.**
(DOCX)

## Acknowledgments

We would like to thank all participants of DEGS1 and of the non-responder survey of DEGS1. Moreover, we would like to thank all members of the joint project AdvanceGender for their contribution in reviewing and discussing the conceptualisation and statistical methodology of this study.

ADVANCE GENDER study group:

University of Bremen, Institute of Public Health and Nursing Research, Department of Social Epidemiology (Gabriele Bolte, Emily Mena), Robert Koch Institute (Alexander Rom-mel, Anke-Christine Saß, Kathleen Pöge, Sarah Strasser), Brandenburg Medical School Theo-dor Fontane, Institute of Social Medicine and Epidemiology (Christine Holmberg, Sibille Merz, Philipp Jaehn).

## Author Contributions

**Conceptualization:** Philipp Jaehn, Christine Holmberg.

**Formal analysis:** Philipp Jaehn.

**Funding acquisition:** Christine Holmberg.

**Methodology:** Philipp Jaehn, Emily Mena.

**Resources:** Robert Hoffmann, Antje Gößwald.

**Supervision:** Christine Holmberg.

**Visualization:** Philipp Jaehn.

**Writing – original draft:** Philipp Jaehn.

**Writing – review & editing:** Philipp Jaehn, Emily Mena, Sibille Merz, Robert Hoffmann, Antje Gößwald, Alexander Rommel, Christine Holmberg.

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
