## [Decision Letter · Decision Letter 0]

1 Jun 2020

PONE-D-20-11368

The intersectionality of non-response in a national health survey in Germany: a multilevel analysis of individual heterogeneity and discriminatory accuracy

PLOS ONE

Dear Dr. Jaehn,

Thank you for submitting your manuscript to PLOS ONE. After careful consideration, we feel that it has merit but does not fully meet PLOS ONE’s publication criteria as it currently stands. Therefore, we invite you to submit a revised version of the manuscript that addresses the points raised during the review process.

A **rebuttal letter** that responds to **EACH** point raised by the academic editor and reviewer(s). You should upload this letter as a separate file labeled 'Response to Reviewers'.A **marked-up copy** of your manuscript that highlights changes made to the original version. You should upload this as a separate file labeled 'Revised Manuscript with Track Changes'.An **unmarked version** of your revised paper without tracked changes. You should upload this as a separate file labeled 'Manuscript'.

We look forward to receiving your revised manuscript.

Kind regards,

Brecht Devleesschauwer

Academic Editor

PLOS ONE

Journal Requirements:

Reviewers' comments:

Reviewer's Responses to Questions

**Comments to the Author**

1. Is the manuscript technically sound, and do the data support the conclusions?

Reviewer #1: Yes

2. Has the statistical analysis been performed appropriately and rigorously? 

Reviewer #1: Yes

3. Have the authors made all data underlying the findings in their manuscript fully available?

Reviewer #1: Yes

4. Is the manuscript presented in an intelligible fashion and written in standard English?

Reviewer #1: Yes

5. Review Comments to the Author

Reviewer #1: This is a worthy study applying an intersectional framework to investigate non-participation in population health surveys. The use of MAIHDA is innovative and clearly suitable. The authors not only focus on the differences between group averages, but they also consider the individual heterogeneity around those averages by interpreting the VPC as a measure of discriminatory accuracy. In my opinion the analytical and conceptual approach used by the authors represent a new and improved way of performing epidemiological analyses. However, I have several comments

R1: In the abstracts, the authors conclude they analyses “do not support targeting specific intersectional strata to achieve more equal study participation”. I think this is right but incomplete. The authors provide a better formulation in page 24, lines 343-345 “Our results indicate that there is no justification to target specific intersectional strata in order to increase response, but that a combination of targeted and population-based measures might be appropriate to achieve more equal representation”.

This concept may appear vague of confuse but it is a relevant issue. It does not mean specific intersectional strata are irrelevant. From an individual/clinical perspective the risk of non-participation in an strata with high average risk may be high. However, a low VPC (low discriminatory accuracy) indicates that focusing ONLY on strata with a low average participation would miss many individuals prone to low participation but belonging to strata with a high average participation. That is, using the strata averages as criteria for intervention may give many false negatives (as well as false positive) if the VPC is low. I suggest, the conclusions in lines 343-34 should also be in the abstracts ( you may add “only”), That is “Our results indicate that there is no justification to target specific intersectional strata ONLY in order to increase response, but that a combination of targeted and population-based measures might be appropriate to achieve more equal representation”.

R2: I think study non-participation by using a survey is interesting but a bit peculiar, as they are non-participants among the non-participants. I would rather use register data to obtain demographic and socioeconomic variables.

In any case you must provide a flow diagram indicating very clearly how many individuals were excluded and the reasons for the exclusion. The reader needs to understand this process very well.

R3: the use of “subjective health” may not fit well intersectionality theory and some sociologists may consider it unsuitable if you do not explain how subjective health condition social location in intrinsic combination with other axes of oppression (see your reference #24 for further explanations).

I would take some distance from feministic intersectionality “theory” and explain the study is inspired on the intersectionality approach concerning socioeconomic variables, but it also adopts a multicategorical/stratified design to provide a better understanding of the heterogeneity in non-participation.

I think the purpose when using MAIHDA is (i) to provide an improved mapping of how non-participation is distributed across the intersectional strata in the population. When doing so, (ii) the method provides reliability weighted strata specific risks and, thereby, can handle strata with small number of individuals. In addition, (iii) intersectional MAIHDA allows to detect strata specific interaction of effects. On top of that, (iv) MAIHDA provides measures of DA that can be used when interpreting the observed differences in non-participation risk across strata.

R3: Regrading the methodology and results I have not mayor comments.

I think you should provide figures with points estimations and confidence intervals rather that only tables.

Please, provide % (n/N) rather than only %

In your sample the problem of strata with very small number of individuals is not actual. However, you should indicate that an advantage of the multilevel analysis is that it uses shrunken residuals so small strata are handle much better than in traditional fixed effects regression.

R4: Finally, the cross-classified MAIHDA allow to separate geographical from socioeconomic influences and evaluate their relative relevance. This is interesting information and you should further discuss this analysis

6. PLOS authors have the option to publish the peer review history of their article (what does this mean?). If published, this will include your full peer review and any attached files.

Reviewer #1: No

---

## [Author Response · Author response to Decision Letter 0]

30 Jun 2020

Dear Prof. Devleesschauwer,

I responded to all comments from the Decision Letter in the cover letter. We responded to all reviewer comments in the Rebuttal Letter.

Kind regards,

Philipp Jaehn

---

## [Decision Letter · Decision Letter 1]

24 Jul 2020

Non-response in a national health survey in Germany: an intersectionality-informed multilevel analysis of individual heterogeneity and discriminatory accuracy.

PONE-D-20-11368R1

Dear Dr. Jaehn,

We’re pleased to inform you that your manuscript has been judged scientifically suitable for publication and will be formally accepted for publication once it meets all outstanding technical requirements.

Kind regards,

Brecht Devleesschauwer

Academic Editor

PLOS ONE

Additional Editor Comments (optional):

Reviewers' comments:

Reviewer's Responses to Questions

**Comments to the Author**

1. If the authors have adequately addressed your comments raised in a previous round of review and you feel that this manuscript is now acceptable for publication, you may indicate that here to bypass the “Comments to the Author” section, enter your conflict of interest statement in the “Confidential to Editor” section, and submit your "Accept" recommendation.

Reviewer #1: All comments have been addressed

2. Is the manuscript technically sound, and do the data support the conclusions?

Reviewer #1: Yes

3. Has the statistical analysis been performed appropriately and rigorously? 

Reviewer #1: Yes

4. Have the authors made all data underlying the findings in their manuscript fully available?

Reviewer #1: No

5. Is the manuscript presented in an intelligible fashion and written in standard English?

Reviewer #1: Yes

6. Review Comments to the Author

Reviewer #1: I think the authors have done a suitable revision. I have not further commenataries. ................

7. PLOS authors have the option to publish the peer review history of their article (what does this mean?). If published, this will include your full peer review and any attached files.

Reviewer #1: No

---

## [Editor Report · Acceptance letter]

30 Jul 2020

PONE-D-20-11368R1 

Non-response in a national health survey in Germany: an intersectionality-informed multilevel analysis of individual heterogeneity and discriminatory accuracy. 

Dear Dr. Jaehn:

I'm pleased to inform you that your manuscript has been deemed suitable for publication in PLOS ONE. Congratulations! Your manuscript is now with our production department. 

Kind regards, 

on behalf of

Prof. Dr. Brecht Devleesschauwer 

Academic Editor

PLOS ONE